# Effects of Yeast β-Glucan Supplementation on Calf Intestinal and Respiratory Health

**DOI:** 10.3390/ani15070997

**Published:** 2025-03-30

**Authors:** Jiamin Wang, Fang Yan, Meng Xiong, Jieru Dong, Wenqian Yang, Xiurong Xu

**Affiliations:** College of Animal Science and Technology, Northwest A&F University, Xianyang 712100, China; jm211219@163.com (J.W.); yfnwafu@163.com (F.Y.); 18108569130@163.com (M.X.); dongjr6688@163.com (J.D.); 3285316728@nwafu.edu.cn (W.Y.)

**Keywords:** holstein cows, yeast β-glucan, diarrhea, bovine respiratory disease, trained immunity

## Abstract

Diarrhea and pneumonia are common diseases in calves worldwide. Based on the theory of trained immunity, this study enabled calves to establish innate immune memory by feeding yeast β-glucan. Pre-stimulation with yeast β-glucan effectively reduced the incidence of diarrhea and pneumonia in Holstein calves from 31 to 60 days of age and enhanced the stress resilience and mucosal defensive ability of calves.

## 1. Introduction

The calf stage is a critical period for dairy cows, which has long-term effects on adult dairy cows [1]. For instance, the average daily gain before weaning was positively correlated with the reproductive rate and milk production during the first lactation period [2]. However, health problems in early life tend to reduce daily gain before weaning, thus limiting the herd’s long-term productive potential [3,4]. Diarrhea and pneumonia are the most common intestinal and respiratory diseases in lactating calves. Severe diarrhea and pneumonia negatively affect growth performance and can even lead to death, which brings great economic losses to the livestock industries [5,6,7]. Infection is the main cause of calf diarrhea and pneumonia. The pathogens are complex and diverse, including common pathogens such as Bovine rotavirus, Bovine viral diarrhea virus, *Escherichia coli*, Cryptosporidium parvum, and *Mannheimia haemolytica* [8,9]. This complexity poses challenges for developing targeted vaccination strategies.

In recent years, the theory of trained immunity, characterized by a significantly enhanced non-specific immune response in trained individuals to respond to both homologous and heterogenous re-infection later [10,11], provides new strategies for the prevention of known or unknown infections in animals, especially in young animals. Immune stimulants capable of inducing trained immunity are primarily pathogens and their attenuated or inactivated strains, as well as their cell wall components [12,13,14]. In addition, it was reported that β-glucans are safe stimulants that can induce innate immunity [15,16]. Only highly purified β-1,3-1, 6-glucans derived from fungi, such as yeast β-glucan, have been proven to induce innate immune memory in cells [17]. Studies have demonstrated that β-glucan, whether administered orally or via injection, effectively induces trained immunity across diverse animal models, enhancing host defense against pathogens. For example, in turbot, intraperitoneal injection of 50 μL of yeast β-glucan (20 mg/mL) significantly reduced mortality after bacterial infection [18]. Similarly, intraperitoneal administration of 0.1 mg of zymosan (a β-glucan-rich cell wall preparation) into mice markedly increased myeloperoxidase activity and prevented peritonitis for up to 5 weeks [19]. In weaned rabbits, intraperitoneal injection of β-glucan (50 mg/kg) 6 and 4 days before weaning significantly reduced post-weaning diarrhea rates [20]. In newborn goats, oral administration of 50 mg/kg body weight of yeast β-glucan on days seven and four before birth was performed. Peripheral blood was collected on day 1 before birth, followed by an LPS challenge. The results showed that β-glucan significantly increased plasma IL-6 levels and the transcription of macrophage markers (CD11b and CD11c), indicating the successful induction of trained immunity. In vitro, β-glucan-treated monocytes showed enhanced respiratory burst activity and inflammatory cytokine secretion upon LPS stimulation [21]. In newborn calves, β-glucan was orally administered at 5 mg/kg body weight on the day of birth and day three. Blood was collected on day seven for monocyte isolation, followed by an in vitro *E. coli* challenge. The results demonstrated that β-glucan upregulated TLR2/NF-κB pathway genes and significantly improved phagocytosis and nitric oxide production while also increasing the consumption and production of glycolytic metabolites [22].

Based on the above-mentioned, we hypothesized that early feeding of yeast β-glucan would induce trained immunity in newborn calves, thereby reducing the incidence of diarrhea and pneumonia during suckling and the early post-weaning period. This study aimed to test this hypothesis by measuring the incidence of diarrhea and pneumonia, growth performance, redox status, immune function, and gut microbiota composition in calves.

## 2. Materials and Methods

### 2.1. Animals and Experimental Design

A total of 44 healthy Holstein heifer calves (22 calves per group), aged 3 days with an average BW (body weight) of 36.18 ± 0.61 kg (mean ± SE), were included in the study. These calves were housed in a naturally ventilated barn with individual shelters measuring 3.0 m × 1.2 m × 1.8 m (length × width × height). The calf’s ID, date, and time of day (morning or evening) were recorded. Each shelter was equipped with water and calf starter buckets. To minimize airborne dust in the housing environment, screened wood shavings with a minimum theoretical length cut of 50 mm were used as bedding. The bedding was refreshed every 7 days to maintain clean and dry pens. The calves were fed 6 L/d of milk, divided into 2 feedings at 06:00 and 16:00, for the first 20 days. Subsequently, they were transitioned to a commercial milk replacer (14% solids, 22.46% CP, 16.20% fat; Sprayfo Azul, Sloten do Brazil Ltd., Deodápolis, MS, Brazil) until the pre-weaning period.

Calves were randomly allocated into two groups based on body weight: the CON (Control) group (*n* = 22) and the PO (Per Os) group (*n* = 22). Calves in the PO group were orally administered a yeast β-glucan solution (0.1 g/mL, 65 mg/kg body weight; purity: 90% on an air-dry basis, obtained from Nanjing Taixin Biotechnology Co., Ltd., Nanjing, China) at 3 and 6 days after birth. In contrast, calves in the CON group received an equivalent volume of physiological saline solution (PSS) at the same time points (Figure 1). Blood samples were randomly collected from 12 calves in each group by jugular venipuncture using BD Vacutainer tubes without anticoagulant (Franklin Lakes, NJ, USA) at 08:00 on days 7 and 30, followed by centrifuge at 3000× *g* for 15 min at 4 °C to isolate serum. Concurrently, fecal samples were obtained through rectal massage into 5 mL frozen tubes. Both serum and fecal samples were promptly frozen in liquid nitrogen and stored at −40 °C for subsequent analysis. At 30 days of age, samples of blood and rectal contents were collected from calves that had not previously had diarrhea.

### 2.2. Growth Performance and Health Recording

Calves were weighed using mechanical scales (ICS-300; Coimma Limited., Rolândia, PR, Brazil) at birth, 30, and 60 days of age, prior to their morning feeding. Before offering fresh starter feed, any feed refusals were removed. Daily individual feed intake was determined by weighing the amounts of starter feed provided, and the amounts refused using a calibrated electronic scale (model PX3000; Pand Iran Co., Isfahan, Iran). The average daily gain (ADG) was obtained by dividing the difference between the measured weight and the birth weight by the date of the interval. Calf-starter intake was computed as the mean amount of feed consumed daily during days 7–30 and days 31–60. Therefore, total dry matter intake = milk replacer intake × 14.3% + calf-starter intake. The average feed conversion ratio (FCR) was determined as the ratio of total DM intake to ADG. Throughout the entire experimental duration, the health status of the calves was monitored daily. Diagnosis of diarrhea was based on fecal consistency, assessed after rectal stimulation to induce defecation. Fecal consistency was scored on a scale of 0 to 3, where 0 denoted normal consistency, 1 indicated semiformal or pasty consistency, 2 represented loose feces, and 3 signified watery feces [23]. A fecal score of ≥2 persisting for more than 2 consecutive days was considered indicative of a bout of diarrhea. BRD assessments were conducted daily by the same trained veterinarian post-morning feeding. Signs such as abnormal nasal discharge, coughing, ear tilt, eye discharge, and elevated rectal temperature (measured using TS-101 Colors Techline digital, Techline São Paulo) were recorded, following the criteria outlined by McGuirk and Peek [24]. A diagnosis of BRD requires the presence of at least 2 categories of abnormal scores. The diarrhea rate (%) for each group was calculated as follows: (number of calves with diarrhea/total number of calves) × 100, while the cumulative rate of diarrhea was determined as previously described [25]. Antimicrobial therapy was administered only when animals exhibited fever or depression symptoms such as recumbency and refusal to consume milk. Calves experiencing a confirmed bout of diarrhea received antimicrobial intervention on the day of diagnosis; sulfamethoxazole and trimethoprim were administered intramuscularly, with the dosage calculated based on BW (1 mL/15 kg; Trissulfim, Ourofino Animal Health), following the protocol outlined by the herd veterinarian. For BRD, florfenicol + flunixin meglumine were administered intramuscularly, with the dosage calculated based on BW (1 mL/15 kg florfenicol; Florkem, Ceva Sante Animale; 1 mL/45 kg flunixin meglumine; Flumax, J.A. Saúde Animal), in accordance with the herd veterinarian’s instructions. Details of medications used, dosages, and duration of treatments were documented for each calf.

### 2.3. The Oxidative Stress Level in Serum

Malonaldehyde (MDA) and Hydroxy free radical scavenging activity in the serum were measured using commercial Enzyme-linked immunosorbent assay (ELISA) kits (Angle Gene Biotechnology, Nanjing, China) according to the manufacturer’s instructions. The product numbers are, in order, ANG-E61182B and YH1250.

### 2.4. Cytokines, Immunoglobulins, and Diamine Oxidase Expression

Cytokines (Interleukin 6 (IL-6), Interleukin-1 beta (IL-1β) and Tumor necrosis factor-α (TNF-α)), Immunoglobulins (IgG (Immunoglobulin G) and IgM (Immunoglobulin M)), and DAO (Diamine oxidase) in the serum were detected using commercial ELISA kits (Jining Biotechnology, Shanghai, China) according to the manufacturer’s instructions. The product numbers are, in order, JN17756, JN19549, JN20140, JN18717, JN17929, and JN24733.

### 2.5. The Levels of Three Mucosal Defense Proteins in Rectal Contents

LYZ (Lysozyme), sIgA (secretory Immunoglobulin A), and defensin in rectal contents were detected using commercial ELISA kits (Jining Biotechnology, Shanghai, China) according to the manufacturer’s instructions. The product numbers are, in order, JN845309, JN18940, and JN84228.

### 2.6. 16S rRNA Gene Sequencing

The DNA extracted from fecal samples of 30-day-old calves was isolated using the CTAB method following the manufacturer’s instructions. For amplification of the V3–V4 region of the 16S rRNA gene, barcoded PCR primers F341 (5′-CCTAYGGGRBGCASCAG-3′) and R806 (5′-GGACTACNNGGGTATCTAAT-3′) were employed. PCR amplification was performed in a total volume of 25 μL reaction mixture containing 25 ng of template DNA, 12.5 μL of PCR Premix, 2.5 μL of each primer, and PCR-grade water to adjust the volume. PCR conditions for amplification consisted of an initial denaturation at 98 °C for 30 s, followed by 32 cycles of denaturation at 98 °C for 10 s, annealing at 54 °C for 30 s, and extension at 72 °C for 45 s, with a final extension step at 72 °C for 10 min. The PCR products were confirmed by electrophoresis on a 2% agarose gel. Throughout the DNA extraction process, ultrapure water was used as a negative control instead of a sample solution to prevent false-positive PCR results. The PCR products were purified by AMPure XT beads (Beckman Coulter Genomics, Danvers, MA, USA) and quantified by Qubit (Invitrogen, Carlsbad, CA, USA). The size and quantity of the amplicon library were evaluated using the Agilent 2100 Bioanalyzer (Agilent, Lexington, MA, USA) and the Library Quantification Kit for Illumina (Kapa Biosciences, Woburn, MA, USA), respectively. Sequencing of the libraries was performed on the NovaSeq PE250 platform following the manufacturer’s protocol provided by LC-Bio. The raw sequencing reads have been deposited in the NCBI Sequence Read Archive under accession number PRJNA1155013. Alpha and beta diversity were assessed by random normalization of the same sequences. Graphical representations were generated using R (version 4.1.2) with the vegan package. To compare microbiome abundances between the CON and PO groups, a Linear Discriminant Analysis Effect Size (LEfSe) analysis was conducted.

### 2.7. Statistical Analyses

Before statistical analysis, all data were recorded and processed in Microsoft Excel 2019 to calculate the standard error of the mean (SEM). All statistical analyses were performed considering the calf as the experimental unit and using SPSS Statistics V27.0.1. The data of growth performance, serum, rectal contents, and intestinal indices in calves were analyzed with independent samples t-tests, while the incidence of diarrhea and BRD were analyzed with the Chi-square test.

For the analysis of microbial data, diversity indices and the relative abundance of phyla between two groups were compared using the nonparametric Wilcoxon rank-sum test. GraphPad Prism 9.5 (GraphPad Software Inc., La Jolla, CA, USA) was used to plot, and all results are expressed as the means ± SEM. Differences were statistically significant at *p* < 0.05.

## 3. Results

### 3.1. Pre-Stimulation with Yeast β-Glucan Induced Inflammatory Response and Oxidative Stress of Calves

The serum level of IL-6 and IL-1β was significantly elevated in the PO group (*p* < 0.01) (Table 1). Additionally, serum MDA content (*p* < 0.001) and hydroxyl free radical scavenging ability (*p* < 0.01) also increased (Table 1). However, there was no significant difference in TNF-α, DAO, IgG, and IgM concentrations between the two groups (*p* > 0.05) (Table 1).

### 3.2. Pre-Stimulation with Yeast β-Glucan Enhanced Concentration of Several Defensive Proteins in Rectal Feces

As shown in Figure 2, the levels of slgA and defensin in the feces of calves at 7 days in the PO group were significantly increased by feeding them yeast β-glucan (*p* < 0.01 or *p* < 0.05), and the LYZ concentration in the rectal feces of the PO group also tended to increase at 7 days of age (*p* = 0.085).

### 3.3. Pre-Stimulation with Yeast β-Glucan Decreased the Incidence of Diarrhea and Pneumonia in Calves

As can be seen from Figure 3, throughout the entire study period, the incidence of diarrhea and pneumonia in calves in the PO group was consistently lower than that in the CON group. Although no statistically significant differences were observed during the periods of 0–30 days of age and the two weeks post-weaning (61–74 days of age), the rate of diarrhea (*p* < 0.05) and pneumonia (*p* < 0.05) in calves was significantly lower in the PO group compared to the CON group during the 31–60 days period. However, as indicated in Table 2, no notable differences were observed in calf starter intake, total DM intake, ADG, or FCR between the CON and PO groups either during the 0–30 or 31–60-day periods. While there was no significant difference in milk intake between the CON and PO groups during the first 30 days, a tendency for increased milk intake was observed in the PO group during days 31–60 (*p* = 0.057). In addition, the PO group tended to have a higher ADG from 0–180 d (*p* = 0.077).

### 3.4. Pre-Stimulation Enhanced the Inflammatory Response and Alleviated Intestinal Damage of Calves at 30 Days Old

The detection results of 30-day-old calves showed that serum IL-6 levels in the PO group was significantly higher than that in the CON group (*p* < 0.01), while the serum DAO level in the PO group was significantly lower than that of the CON group (*p* < 0.001) (Figure 4A,C). The serum concentrations of IL-1β, TNF-α, IgG, and IgM are not significantly different from those in the CON group (Figure 4A,B). In addition, at 30 days of age, the slgA level in rectal feces of pre-stimulated calves was also significantly higher than that of the CON calves (*p* < 0.05), but there was no significant difference in fecal defensin and LYZ levels between the two groups (*p* > 0.05) (Figure 4D).

### 3.5. Pre-Stimulation with Yeast β-Glucan Alleviated the Oxidative Stress of Calves Later

The oxidative stress status in 30-day-old calves was also analyzed. As shown in Figure 5, the level of serum malondialdehyde (MDA) of pre-stimulated calves was significantly lower (*p* < 0.05), and their hydroxyl radical scavenging capacity was markedly higher than that of the CON calves (*p* < 0.05).

### 3.6. Pre-Stimulation with Yeast β-Glucan Altered the Rectal Bacterial Community of Calves at 30 Days Old

For α diversity, we used chao1 and observed species to reflect the microbial richness and the Shannon and Simpson index to reflect microbial diversity. As shown in Figure 6A, there was no significant difference in richness and diversity. Analysis of β diversity based on PCoA (principal coordinates analysis) of unweighted unifrac distance showed different clustering distributions (*p* = 0.001), as demonstrated in Figure 6B. Furthermore, the individual samples of the PO group were clustered together on the PCoA graph, while the individual samples of the CON group were dispersed. The relative abundance and composition of the top 10 at the phylum level are shown in Figure 6C. The fecal bacterial community was dominated by Firmicutes and Bacteroidota, followed by Actinobacteriota. For further assessment of the differences in fecal microbiota among the samples of the two groups, LDA (Linear Discriminant Analysis) was carried out for fecal microorganisms at different taxonomic levels with LDA > 3.0 (*p* < 0.05) as the critical value. Some genera that enriched in the CON group included *Christensenellaceae_R_7_group*, *Pedobacter*, *Erysipelatoclostridium*, *Intestinimonas*, and *Fusobacterium*. Eleven genera were significantly enriched in the PO group, consisting of *Subdoligranulum*, *Megamonas*, *Enterococcus*, *Dorea*, *Clostridium*_*sensu*_*stricto*_*2*, *Muribaculaceae*_*unclassified*, *Allobaculum*, *Clostridium*_*sensu*_*stricto*_*13*, *Ligilactobacillus*, *Brevundimonas*, and *Lachnospiraceae*_*NK4A136*_*group* (Figure 6D).

## 4. Discussion

Newborn calves are highly susceptible to diarrhea or BRD due to their immature immune system, leading to high mortality and long-term health issues [3]. While vaccines are effective against specific pathogens, the complexity of causative agents complicates prevention [26,27]. The concept of trained immunity, which enhances innate immune responses against diverse pathogens, offers a promising alternative. Studies have shown that innate immune stimulants improve survival rates in animals exposed to infections [28,29,30], highlighting their potential for preventing complex diseases in young animals like calves.

Yeast β-glucan is widely used as a prebiotic in livestock. However, increasing numbers of studies have proved that yeast β-glucan can be used as an immune stimulant to induce trained immunity. Although its role in preventing diarrhea and pneumonia in calves has not been extensively studied, in vitro and in vivo experiments have confirmed its ability to induce trained immunity in bovine monocytes [22]. Unlike previous studies that primarily used injectable stimulants, our study employed oral administration to minimize stress and mimic natural exposure. The aim of this study was to investigate whether early oral administration of yeast β-glucan could effectively prevent diarrhea and respiratory disease in calves.

According to the theory of trained immunity, the inflammatory response triggered by pre-stimulation is essential for establishing immune memory [31]. As supposed, feeding yeast β-glucan twice induced an obvious inflammatory response in calves, and serum IL-6 and IL-1β levels increased significantly 24 h after the second supplementation with yeast β-glucan. Of course, studies conducted on mice and rats have demonstrated that the induced inflammatory response is typically transient, subsiding within two to three days [32,33]. Importantly, stimulation with yeast β-glucan did not increase serum DAO levels in calves, suggesting that the inflammatory response induced by oral administration of yeast β-glucan did not cause investigated damage to the intestinal tissue of calves. More importantly, this response did not increase serum DAO levels, indicating no intestinal tissue damage. As expected, the incidence of diarrhea and pneumonia of the pre-stimulated calves at 30–60 days of age were significantly lower than those in the CON group, indicating that early-stage yeast β-glucan administration provided protective effects against subsequent systemic infections.

The defensive effects observed were consistent with the induction of trained immunity. Classical studies of trained immunity are divided into three stages: pre-stimulation, resting period, and re-stimulation. However, in feeding experiments, it is challenging to predict when each calf will encounter subsequent re-infection after pre-stimulation. However, all calves in our experiment experienced a combination of stresses before and after 30 days of age, including changes in milk, sharp temperature drops, and taking off vests. Calves are particularly susceptible to cold conditions due to their low body surface area-to-weight ratio and thin layers of skin and subcutaneous fat [34]. Additionally, their underdeveloped rumens prevent them from generating sufficient heat through fermentation. Stress has been proven to increase the risk of pathogen infection in calves [35,36]. Therefore, the combined stresses can be considered as the re-stimulation phase in classical trained immunity studies. The typical phenotype of the trained immunity re-stimulation phase includes an enhanced defense response of innate immune cells and remission of tissue damage, but it is accompanied by increased expression of certain pro-inflammatory factors such as IL-6, IL-1β, and TNF-α [37,38]. In the present study, the levels of intestinal damage marker DAO and oxidative stress marker MDA in the serum of pre-stimulated calves at 30 days of age were significantly decreased, while the levels of IL-6 in the serum were significantly higher than those in the CON group. The above evidence is consistent with the phenotype of the re-stimulation phase in the classical studies of trained immunity.

Although we did not directly measure mucosal defense proteins in intestinal tissue, higher levels of sIgA in the rectal contents of the PO group at 30 days of age indicated enhanced mucosal immunity. Our previous study showed that the sIgA in the intestinal mucosa of trained rats was higher than that of untrained ones when they all were challenged with a heterologous pathogen at a later stage [39]. It has been confirmed that intestinal innate lymphoid cells are also subject to training to enhance the immune response to re-stimulation [40]. Whether the enhanced sIgA expression was associated with a possible enhanced immune response of gut-associated innate immune cells or intestinal epithelial cells was unclear. However, our results showed that previous stimulation enhanced the immune response of intestinal mucosa in response to comprehensive stress at a later stage, which was consistent with a previous report [41].

Beta diversity analysis of the rectal bacterial community showed that pre-stimulation resulted in more homogeneous and clustered individuals in the PO group, indicating greater microbial stability.

LEfSe analysis further showed a significant decrease in *Christensenellaceae_R_7_group* and *Fusobacterium*, which are associated with metabolic disorders and increased diarrhea severity [42,43], and a significant increase in *Subdoligranulum* and *Dorea*, which are linked to healthy gut microbiota [44,45]. In addition, the PO group is enriched with bacteria capable of producing short-chain fatty acids, particularly butyrate, including *Muribaculaceae*, *Lachnospiraceae*, and *Dorea*. Existing research has shown that the gut microbiota not only influences the development of host immune cells but also regulates their differentiation and function through its components and derived metabolites [46]. For example, SCFAs, such as butyrate, play a significant role in modulating the host immune system and enhancing intestinal barrier function, thereby supporting immune homeostasis [47,48]. These results suggested that fed yeast β-glucan has a beneficial effect on calf microflora structure.

Studies have confirmed that the daily addition of β-glucan as a prebiotic to a base diet can improve animal performance [49,50]. However, our present study showed that although pre-stimulation was beneficial to the intestinal and lung health of calves, it had no significant positive effects on the body weight, average feed intake, average daily gain, and feed-to-gain ratio of calves during the suckling period. We speculated that the reasons may be involved in the following aspects: Firstly, as an immune stimulant, the total amount of glucan consumed during the whole lactation period is far lower than the total amount as a daily supplement of prebiotics. Secondly, compared with the CON group, the immune response of the pre-stimulated calves was induced during the stimulation period, and the immune level of the calves was also higher when they were affected by the combined stress at about 30 days of age. This meant that during both phases, the pre-stimulated calves required greater energy expenditure for the immune response. Thirdly, the two feeding methods have different regulatory mechanisms for the gut microbiota of calves, so the resulting gut microbiota composition will also differ, and the microbiota changes caused by the addition of prebiotics may have a greater impact on growth performance. These combined factors may lead to the difference in the effects of oral yeast β-glucan as a stimulant on the growth performance of calves during the lactation stage compared with the addition of yeast β-glucan as a prebiotic in the feed. It is noting that we later tracked the body weight of calves at six months of age in the two groups, and the analysis results showed that the daily gain of calves at six months of age in the pre-stimulation group was higher than that in the CON group, indicating the subsequent beneficial effect of previous stimulation with yeast β-glucan. Of course, though early pre-stimulation with yeast β-glucan decreased the diarrhea rate and BRD before calves weaning and increased their body weight at 6 months, whether these positive effects have a positive impact on the production performance of adult calves needs follow-up research.

## 5. Conclusions

In conclusion, this study demonstrates that feeding yeast β-glucan twice effectively trains calves to cope with later stress, significantly reduces the incidence of diarrhea and pneumonia from day 31 to day 60 in Holstein calves, and improves intestinal health. In addition, the results of the relevant indexes indicate that these benefits are related to the induction of trained immunity in the calves by pre-stimulation. However, long-term follow-up trials are needed to investigate the effects of early trained immunity in calves on their adult performance.

## Figures and Tables

**Figure 1 animals-15-00997-f001:**
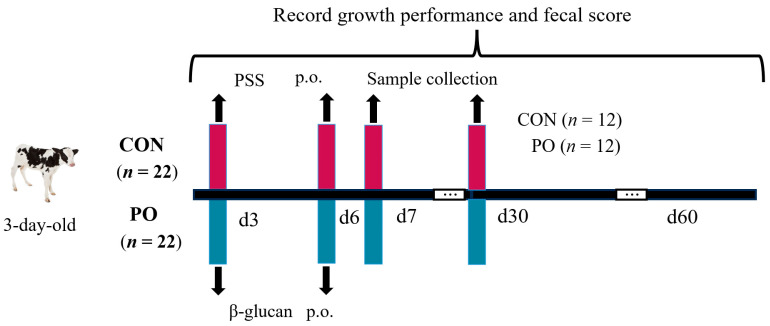
Schematic diagram of the experimental procedure. Calves were treated with yeast β-glucan or PSS on days 3 and 6, respectively, and samples were collected on days 7 and 30. Growth performance and fecal scores were recorded until two weeks after weaning. PSS, Physiological saline solution.

**Figure 2 animals-15-00997-f002:**
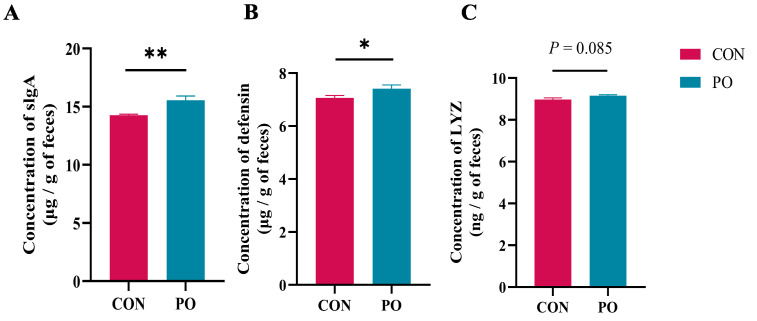
Levels of three defense proteins in feces of calves at 24 h after the second administration (*n* = 12/group). (**A**) sIgA, (**B**) defensin, and (**C**) LYZ. All data are presented as mean ± SEM. Asterisks indicate significant differences between the PO group and the CON group (* *p* < 0.05; ** *p* < 0.01). PO, Per Os group; CON, Control group; sIgA, secretory Immunoglobulin A; LYZ, Lysozyme.

**Figure 3 animals-15-00997-f003:**
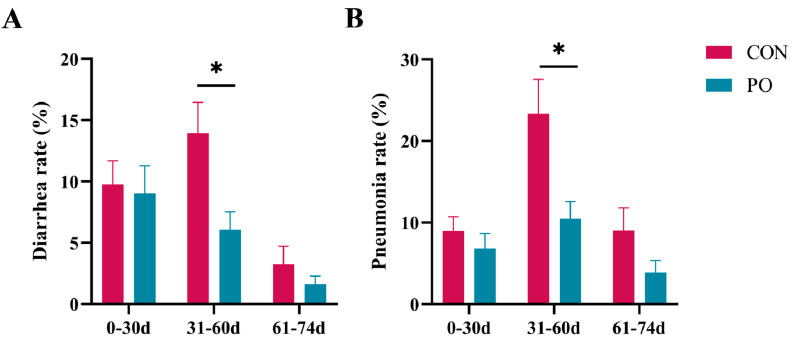
The effect of pre-stimulation with yeast β-glucan on the incidence of diarrhea (**A**) and pneumonia (**B**) in calves later (*n* = 22/group). All data are presented as mean ± SEM. SEM, standard error of the means. PO, Per Os group; CON, Control group. Asterisks indicate significant differences between the PO group and the CON group (* *p* < 0.05).

**Figure 4 animals-15-00997-f004:**
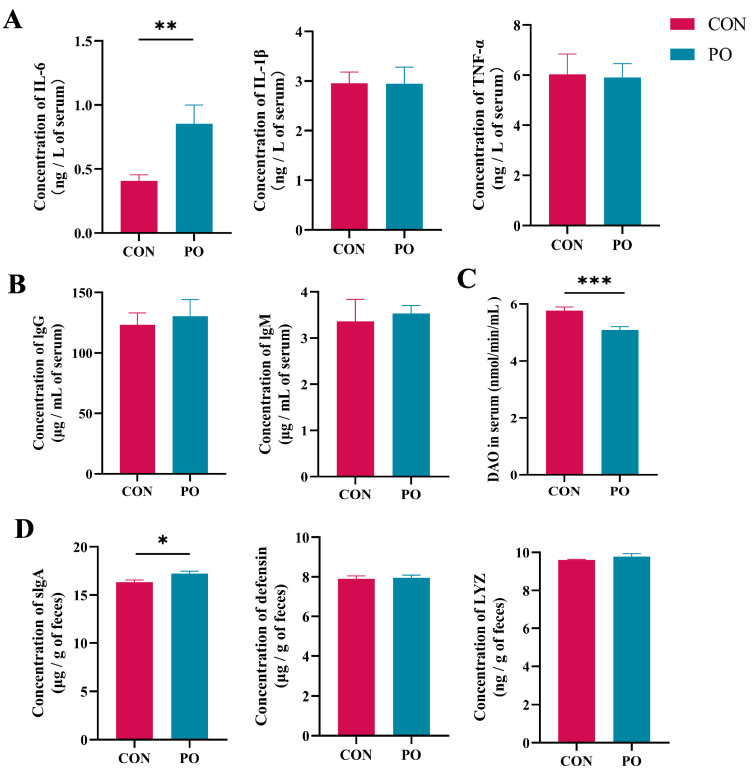
The effect of pre-stimulation with yeast β-glucan on the levels of cytokines, immunoglobulins, DAO in serum, and defense proteins in feces of calves at 30 days of age (*n* = 12/group). (**A**) Level of three inflammatory cytokines in serum of calves. (**B**) Level of two immunoglobulins in serum of calves. (**C**) Levels of DAO in serum of calves. (**D**) Levels of three defense proteins in feces of calves at 30 days of age. All data are expressed as the means ± SEM. SEM, standard error of the means. PO, Per Os group; CON, Control group. Asterisks indicate significant differences between the PO group and the CON group (* *p* < 0.05; ** *p* < 0.01, *** *p* < 0.001).

**Figure 5 animals-15-00997-f005:**
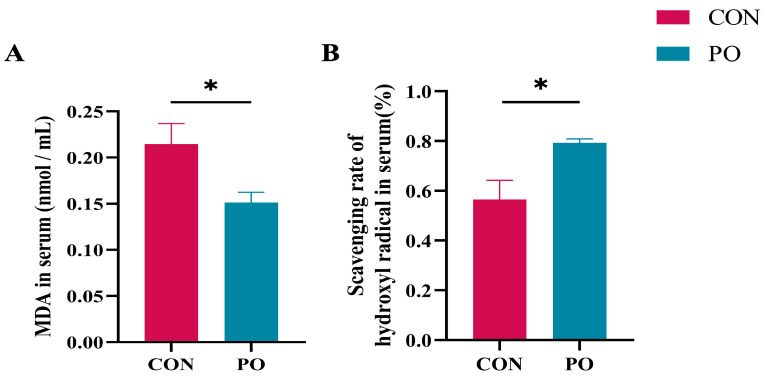
The effect of pre-stimulation with yeast β-glucans on oxidative stress level of calves at 30 days of age. (**A**) Levels of MDA in serum of calves (*n* = 12/group). (**B**) Levels of hydroxyl radical scavenging capacity in serum of calves. All data are presented as mean ± SEM. SEM, standard error of the means. PO, Per Os group; CON, Control group. Asterisks indicate significant differences between the PO group and the CON group (* *p* < 0.05).

**Figure 6 animals-15-00997-f006:**
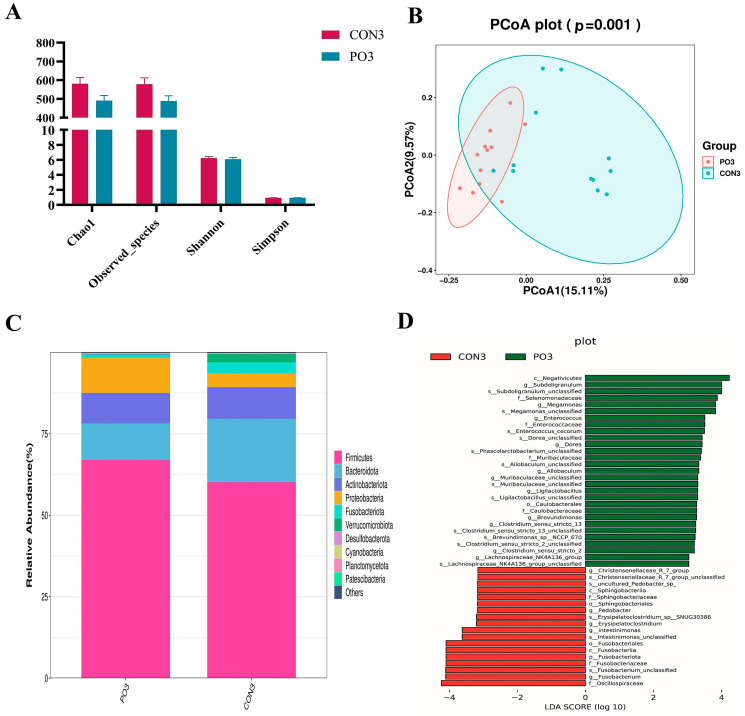
The effect of pre-stimulation with yeast β-glucan on the rectal bacterial community of calves at 30 days of age (*n* = 12/group). (**A**) Analysis of alpha diversity of fecal flora in two groups. (**B**) PCoA diagram of Beta diversity in fecal flora between the two groups. (**C**) Relative abundance of species at phylum level of fecal flora between the two groups. (**D**) Effect sizes were analyzed by LDA of fecal microbiota (LDA > 3.0, *p* < 0.05). PO, Per Os group; CON, Control group.

**Table 1 animals-15-00997-t001:** Determination of immune factors in calf serum 24 h after the second feeding (*n* = 22 calves/group).

Items	Experimental Groups		SEM	*p*-Value
	CON	PO		
IL-6, ng/L	0.212	0.518	0.088	0.008
IL-1β, ng/L	2.149	2.996	0.154	0.003
TNF-α, ng/L	4.037	4.002	0.361	0.698
IgG, μg/mL	99.002	118.057	6.367	0.138
IgM, μg/mL	2.937	3.523	0.182	0.110
Scavengiing rate of hydroxyl radical, %	0.739	0.965	0.039	0.002
MDA, nmol/mL	0.241	0.523	0.030	<0.001
DAO, nmol/min/mL	5.916	5.937	0.315	0.974

Note: All data are expressed as the means ± SEM. SEM, standard error of the means; PO, Per Os group; CON, Control group; IL-6, Interleukin 6; IL-1β, Interleukin-1 beta; TNF-α, Tumor necrosis factor-α; IgG, Immunoglobulin G; IgM, Immunoglobulin M; MDA, Malonaldehyde; DAO, Diamine oxidase.

**Table 2 animals-15-00997-t002:** Effects of β-glucan stimulation on growth performance in calves (*n* = 22 calves/group).

Items	Experimental Groups		SEM	*p*-Value
	CON	PO		
0–30 d				
Milk replacer intake, L/d	8.524	8.618	0.056	0.408
Calf-starter intake, kg/d	0.035	0.039	0.006	0.727
Total DM intake, kg/d	1.254	1.270	0.010	0.475
ADG, kg/d	0.983	0.944	0.037	0.602
FCR	1.352	1.433	0.026	0.506
31–60 d				
Milk replacer intake, L/d	9.262	9.476	0.057	0.057
Calf-starter intake, kg/d	0.126	0.105	0.011	0.323
Total DM intake, kg/d	1.451	1.460	0.012	0.708
ADG, kg/d	0.982	1.028	0.035	0.511
FCR	1.527	1.496	0.035	0.765
0–60 d				
Milk replacer intake, L/d	8.893	8.922	0.079	0.518
Calf-starter intake, kg/d	0165	0.143	0.015	0.494
Total DM intake, kg/d	1.436	1.440	0.016	0.927
ADG, kg/d	0.982	1.001	0.009	0.324
FCR	1.465	1.439	0.015	0.413
0–180 d				
ADG, kg/d	1.042	1.086	0.012	0.077

Note: All data are expressed as the means ± SEM. SEM, standard error of the means; PO, Per Os group; CON, Control group; DM, Dry Matter; ADG, Average Daily Gain; FCR, Feed Conversion Ratio.

## Data Availability

The raw data supporting the conclusions of this article will be made available by the authors on request.

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
