# Peer review of "Effects of Yeast β-Glucan Supplementation on Calf Intestinal and Respiratory Health"

_animals, 2025, doi:10.3390/ani15070997_

Round 1
Reviewer 1 Report
Comments and Suggestions for Authors
Manuscript details:
Journal: Animals
Manuscript ID: animals-3529901
Type of manuscript: Article
Title: Evaluation of the Effects of Feeding Yeast β-glucan on the Intestinal and Respiratory Health of Calves
Authors: Jiamin Wang, Fang Yan, Meng Xiong, Jieru Dong, Wenqian Yang, Xiurong Xu *
The authors investigated the effects of feeding yeast β-glucan on the intestinal and respiratory health of calves during suckling period. Newborn Holstein calves in the PO group were fed yeast β-glucan solution (0.1 g/mL, 65 mg/kg body weight, n = 22) at 3 and 6 days of age, respectively. They found that pre-stimulation through feeding yeast β-glucan also improved the health of calves during suckling period, and provided an experimental and theoretical basis for the prevention of intestinal and respiratory infections in calves by inducing trained immunity. While this work is valuable, several concerns were raised throughout the text and need to be clarified by the authors before further consideration. Please refer to the comments in the PDF file and revise accordingly before further consideration.

Reviewer 2 Report
Comments and Suggestions for Authors
Comments and Suggestions for Authors
After reviewing the manuscript entitled “Evaluation of the Effects of Feeding Yeast β-glucan on the Intestinal and Respiratory Health of Calves”, the following suggestions were made it. The manuscript contains novel and interesting information on the use of yeast β-glucans in calves' respiratory and intestinal health. However, a major revision is required before the manuscript can be considered for publication in Animals. The individual corrections are shown below:
Abstract
Lines 21-24: Authors should clearly describe the PO and CON groups before using these abbreviations for the first time. Also, the number of replicates (calves) used within each treatment evaluated should be specified. The experimental design and average weight (with standard deviation) of the calves used should be specified.
Lines 25-36: The description of the results is not clear and should be improved by adding the specific significance value observed in the different response variables. Also, the authors misused capital letters because capital letters should only be used at the beginning of a new sentence or paragraph (after a period).
Lines 37-40: The information in the conclusion lines is adequate. However, these lines need to be rewritten because, in their current form, they seem more like a summary description of the results. Also, this section should be written in the present tense.
Introduction
Line 53: Although the pathogens that cause diarrhea and pneumonia are complex and varied, the authors should add at least a couple of the main bacteria causing these diseases in dairy calves.
Lines 60-67: Background on yeast β-glucans used as a treatment is not sufficient. β-glucans are only one of several types of prebiotics currently available for cattle feed. Although β-glucans are one of the most commonly used prebiotics, there are also other prebiotics, such as mannan-oligosaccharides, fructo-oligosaccharides, and xylooligosaccharides that are widely used as additives for dairy calves and other farm animals worldwide. Therefore, the authors should add a few lines justifying the use of β-glucans instead of other common prebiotics. This justification should highlight biological, economic, and management aspects that favor β-glucans. On the other hand, the authors should review and cite in this introductory paragraph at least five scientific articles that evaluated β-glucans in dairy calves with positive effects (it is possible to use data from other non-ruminants to expand the background). When adding information from these β-glucan articles, authors should specify the doses and route of administration of β-glucans, the experimental periods, and the benefits observed in the species evaluated.
Line 69: After correcting the information requested in previous comments, authors should add a clear hypothesis of the expected results before the start of the experimental phase (this serves as support to justify the relevance of the current study).
Line 70: Please delete this phrase “which is generally supplemented daily as a prebiotic”.
Lines 73-76: These lines should be joined with lines 69-72 to describe the objective of the study in a single sentence and more clearly.
Line 77: Please delete this phrase “Our study may provide a new and effective measure to prevent intestinal and respiratory diseases of calve”.
Material and methods
Lines 82-85: Please specify the number of replicates within each treatment.
Lines 93-95: The difference between the control and the experimental treatment should be described in detail. Please describe the treatments in detail in this section.
Lines 100: It should be specified whether the tubes used contained anticoagulant.
Lines 111-188: The description of all these materials and methods is well-written and organized and contains sufficient information to make it replicable.
Lines 191-192: Please specify what type of processing is performed using Microsoft Excel. The rest of the statistical analysis information is correct and well-written.
Results
Lines 204-205: It is not necessary to repeat the procedures performed again. Please delete these lines and describe the results tables directly.
Lines 213: The table footer should describe the meaning of the abbreviations CON and PO shown within the results box.
Lines 230-233: The meaning of the abbreviations CON and PO shown within the figure should be described in the caption of Figure 2.
Lines 236-237: There is no need to repeat the objective of the procedures performed again. Please delete these lines and describe the results directly in the table.
Lines 251-254: The meaning of the abbreviations CON and PO shown within the figure should be described in the caption for Figure 3. This correction should be applied to all figures in the manuscript.
Lines 256-257: The meaning of the abbreviations CON and PO shown in the results table should be described in the footer of Table 2. Please apply this correction to all results tables in the manuscript.
Lines 296-297: Please delete these lines.
The remainder of the results section is well written and organized, and contains an adequate number of tables and figures.
Discussion
Lines 322-337: The authors provide an excessive amount of background in these lines. Therefore, this paragraph should be shortened. The discussion section explains the results obtained and compares them with those previously obtained by other authors. However, in its current form, the authors provide a discussion that seems more like a literature review.
Lines 338-449: Overall, the entire discussion section is too ambiguous. β-glucan prebiotics have a wide variety of mechanisms of action, which the authors did not mention. The discussion of all response variables should be improved by using the mechanisms of action of yeast β-glucans to explain the observed results on calf health parameters. There is probably little information on β-glucans in calves; however, β-glucans have been widely evaluated in other farm animal species, and the authors could use that information. In discussing and contrasting the results of the present study with previous literature, the authors should specify the doses of β-glucan and the experimental periods so that the reader can understand some of the main factors contributing to the variability of results between studies.
Conclusions
Lines 451-458: The information in these lines is adequate. However, it should be rewritten in the present tense. Also, the authors' perspectives, such as lines 456-458, should be removed.
Comments on the Quality of English LanguageThe English language require some improvements
Round 2
Reviewer 1 Report
Comments and Suggestions for Authors
Present form can be accepted.
Reviewer 2 Report
Comments and Suggestions for Authors
Comments and Suggestions for Authors
After reviewing the manuscript entitled “Effects of Yeast β-glucan Supplementation on Calf Intestinal and Respiratory Health”, the following suggestions were made it. The authors have made a great effort to respond to my comments and satisfactorily made all the requested corrections. Therefore, I have no additional suggestions and believe the manuscript can be accepted for publication in its current form.